# Frontal beta-theta network during REM sleep

**Sujith Vijayan[1,2]\*, Kyle Q Lepage[1], Nancy J Kopell[1], Sydney S Cash[2]**

[1]Department of Mathematics and Statistics, Boston University, Boston, United States; [2]Department of Neurology, Harvard Medical School and Massachusetts General Hospital, Boston, United States

**Abstract** We lack detailed knowledge about the spatio-temporal physiological signatures of REM sleep, especially in humans. By analyzing intracranial electrode data from humans, we demonstrate for the first time that there are prominent beta (15–35 Hz) and theta (4–8 Hz) oscillations in both the anterior cingulate cortex (ACC) and the DLPFC during REM sleep. We further show that these theta and beta activities in the ACC and the DLPFC, two relatively distant but reciprocally connected regions, are coherent. These findings suggest that, counter to current prevailing thought, the DLPFC is active during REM sleep and likely interacting with other areas. Since the DLPFC and the ACC are implicated in memory and emotional regulation, and the ACC has motor areas and is thought to be important for error detection, the dialogue between these two areas could play a role in the regulation of emotions and in procedural motor and emotional memory consolidation.

\*For correspondence: svijayan9@gmail.com

**Competing interests:** The authors declare that no competing interests exist.

## Introduction

A night's sleep consists of periods of rapid eye movement (REM) sleep and periods of non rapid eye movement (NREM) sleep. We have a relatively detailed picture of the spatio-temporal pattern of activity that occurs during NREM sleep, and this has allowed detailed hypotheses concerning the circuits and mechanisms at play during NREM sleep. For example, the transfer of memories from the hippocampus to the neocortex during NREM sleep is thought to be mediated by the coordination of oscillatory activity seen in the local field potentials, namely by the interaction of hippocampal ripples (80–250 Hz) with neocortical spindles (11–16 Hz) and slow (<1 Hz) oscillations (*Battaglia et al., 2004*; *Ji and Wilson, 2007*; *Lee and Wilson, 2002*; *Mölle et al., 2006*; *Siapas and Wilson, 1998*; *Sirota et al., 2003*; *Skaggs and McNaughton, 1996*; *Staresina et al., 2015*; *Wilson and McNaughton, 1994*).

The spatio-temporal patterns of activity during REM sleep are not as well characterized. This may be due in part to the fact that the activity patterns seen during REM sleep do not easily lend themselves to neat description like those seen during NREM sleep. During REM sleep field potential recordings are not dominated by prominent oscillations; rather they are irregular and of low voltage, similar to recordings during awake behavior (*Rasch and Born, 2013*). During awake activity, the oscillations that emerge during cognitive tasks are thought to play a key role in aspects of cognition such as working memory and communication between brain areas (*Başar et al., 2001*). Thus a detailed understanding of the spatio-temporal characteristics of emergent oscillations during REM sleep may shed light on which areas are interacting during REM sleep, how dreams are generated, and how some of the putative functions of REM sleep, such as memory consolidation, are carried out. Little is known about oscillatory activity during REM sleep with the notable exceptions of theta oscillations in the rodent hippocampus (*Buzsáki, 2002*; *Louie and Wilson, 2001*; *Montgomery et al., 2008*), gamma oscillations in the neocortex (*Castro et al., 2013*; *Llinás and*

**eLife digest** Over the course of a night we cycle through several different stages of sleep. During one of these stages, our eyes move rapidly from side to side behind our closed eyelids. This movement gives this stage its name: rapid eye movement sleep, or REM sleep for short. Most other muscles are paralyzed during REM sleep, possibly to prevent us from acting out the vivid dreams that also occur during this stage of sleep. But despite the distinctive properties of REM sleep, relatively little is known about about why we need it or how the brain generates it.

Vijayan et al. have now obtained new insights into the brain activity that underlies REM sleep by recording from the brains of human patients with epilepsy. The patients all had electrodes temporarily inserted into their brains to help neurologists identify the area of the brain that was responsible for their seizures. By recording from these electrodes overnight, Vijayan et al. were able to study the activity of individual brain regions while the patients slept.

Analysis of the recordings revealed rhythmic waves of neuronal activity in areas at the front of the brain during REM sleep. Two types of brain waves dominated: theta waves, which are relatively slow waves with a frequency of 4–8 cycles per second (Hertz), and beta waves, which are faster with a frequency of 15–35 Hertz. These theta and beta waves were especially pronounced in two subregions of the frontal lobe of the brain, called the dorsolateral prefrontal cortex (DLPFC) and the anterior cingulate cortex (ACC).

The discovery of prominent rhythmic activity in the DLPFC was unexpected. This is because previous studies had shown that this region, which is involved in decision-making and planning, was relatively inactive during REM sleep. Indeed it had been suggested that the limited activity of the DLPFC subregion might be responsible for the often bizarre and illogical nature of our dreams. Instead, Vijayan et al. showed that the ACC and the DLPFC coordinate their activity during REM sleep. The next challenge is to find out whether this dual activity helps support other roles that the two regions share in common, such as the strengthening of memories and the regulation of emotions.

Ribary, 1993; Steriade et al., 1996), and pontine-geniculate-occipital (PGO) waves (Calvo and Fernández-Guardiola, 1984; Datta, 1997; Datta and Hobson, 1994; Steriade et al., 1989).

We decided to focus on the oscillatory dynamics of the frontal cortices during REM sleep based on imaging studies and REM sleep's putative functional roles. Imaging studies suggest that activity patterns in certain regions of the frontal cortices are markedly different during REM sleep than during awake activity, with some areas (e.g., the anterior cingulate (ACC)) being more active and some (e.g., the dorsolateral prefrontal cortex (DLPFC)) less active (Braun et al., 1997; Maquet et al., 1996). These differences in activity patterns suggest that important REM sleep processes might be taking place in the frontal cortices (Braun et al., 1997; Maquet et al., 1996; Muzur et al., 2002). At a functional level, REM sleep has been hypothesized to play a role in such processes as the consolidation of emotional and procedural motor memories (Fischer et al., 2002; Gilson et al., 2016; Nishida et al., 2009; Nitsche et al., 2010; Rasch and Born, 2013; Smith, 2001) and the regulation of emotions. Since the frontal cortices serve memory, emotional regulation (Buhle et al., 2014; Etkin et al., 2011, 2006), and motor functions, they make good candidate areas for the mediation of procedural motor memory and emotional memory consolidation as well as emotional regulation during REM sleep.

We show that the frontal cortices are dominated by bursts of beta activity and theta oscillations, in particular in the ACC and the DLPFC. The prominent oscillatory activity in the DLPFC is especially surprising since current thinking, based on imaging studies (Braun et al., 1997; Maquet et al., 2000; Muzur et al., 2002), is that the DLPFC is relatively quiet during REM sleep in comparison with awake periods, and it has been postulated that the relative quiescence of this 'executive' structure might be responsible for the bizarre nature of our dreams. We also show the novel result that beta bursts and theta oscillations during REM sleep are coherent between the DLPFC and the ACC, two relatively distant structures. This finding is especially intriguing since the ACC and the DLPFC are both implicated in memory and emotional processing (Buhle et al., 2014; Etkin et al., 2011,

*2006*; *Goldman-Rakic, 1995*), and the ACC is interconnected with motor areas (*Bates and Goldman-Rakic, 1993*; *Dum and Strick, 2002*, *1991*; *Hatanaka et al., 2003*; *Morecraft and Van Hoesen, 1993*, *1992*; *Picard and Strick, 1996*), is thought to have motor regions itself (*Paus, 2001*), and is thought to be important in error detection (*Holroyd and Yeung, 2011*; *Paus, 2001*). Therefore the rhythmic activity we observe in the frontal cortices may play a role in some of the hypothesized functions of REM sleep such as the consolidation of emotional and procedural motor memories and the regulation of emotions.

## Results

We examined overnight sleep recordings of five patients with intractable epilepsy undergoing invasive recordings to localize the focus of their seizures. Patient information is in *Table 1*. We used electroencephalography (EEG), electromyography (EMG), and electrooculography (EOG) to score the sleep recordings in 30 s epochs as either N1, N2, N3 (NREM stages), REM, or Awake using standard criteria (*Iber et al., 2007*). According to these criteria, REM periods are identified by low EMG power (the lowest of a night's sleep), irregular EEG of low voltage, and conjugate eye movements (*Figure 1a,b*). Identified REM periods were selected for further analysis.

All patients were monitored using intracranial depth electrodes whose distal-most contacts targeted the medial temporal and frontal regions (*Figure 1c,d*). Many of the frontal electrodes showed a marked change in the spectral content during REM sleep onset. In particular, REM sleep onset resulted in the appearance of prominent bursts of beta activity. *Figure 2a* depicts the spectrogram from an exemplar frontal electrode. The activity shown here and in other figures (and the activity included in all analyses) was detected by bipolar derivations from two neighboring contacts; thus this physiological activity is relatively local in nature. As REM sleep begins there is a change in the spectral content; beta bursts appear and then occur throughout the REM sleep episode. To get an impression of the global occurrence of the beta bursts we identified periods during which there was a prominent beta activity on a given electrode (three standard deviations above the median power and at least two cycles in length). This analysis suggested that the beta bursts indeed tended to occur in the frontal electrodes after onset and throughout the REM sleep episode, but not in temporal lobe electrode leads (*Figure 2b*). Furthermore, the beta bursts seemed to occur in close proximity to one another across the frontal electrodes.

To determine where exactly the beta bursts were prominent within the frontal cortices, and to determine the prominent spectral features of the frontal electrodes, we focused on specific regions within the frontal cortices. We first examined the activity of contacts in the DLPFC (See Methods for electrode localization details), an area that imaging studies suggest is relatively inactive during REM sleep relative to awake periods (*Braun et al., 1997*; *Maquet et al., 1996*; *Muzur et al., 2002*). Surprisingly, the DLPFC contacts showed bursts of oscillatory activity throughout each period of REM sleep, with activity occurring most prevalently in the beta and theta bands (*Figure 3a,b*). All of the DLPFC electrode contacts (15/15) showed significant theta activity (t test, p<0.05/15, Bonferroni corrected for number of electrodes (15), median 5.2 Hz, *Figure 3c,d*), while ~87% (13/15) of the

**Table 1.** Patient Information. Each row provides a patient's demographic information and diagnostic information about their epilepsy.

| Subject | Gender | Age | Handedness | Diagnosis | Imaging |
|---|---|---|---|---|---|
| 1 | F | 45 | R | Bilateral temporal lobe epilepsy | MRI-normal<br>PET-hypometabolism right side |
| 2 | F | 55 | R | Bilateral temporal lobe epilepsy | MRI-mild T2 hyperintensity and volume loss (right greater than left) in the mesial temporal lobes |
| 3 | F | 45 | R | Multifocal temporal parietal-occipital epilepsy | MRI- multifocal FlAIR abnormalities - second lymphoma? |
| 4 | F | 42 | Ambidextrous | Multifocal epilepsy with involvement of bilateral temporal lobes | MRI- nonspecific T2 hyperintensities<br>PET-without clear lesion |
| 5 | M | 53 | Left | Focal epilepsy etiology and localization unknown | MRI- normal<br>PET- normal |

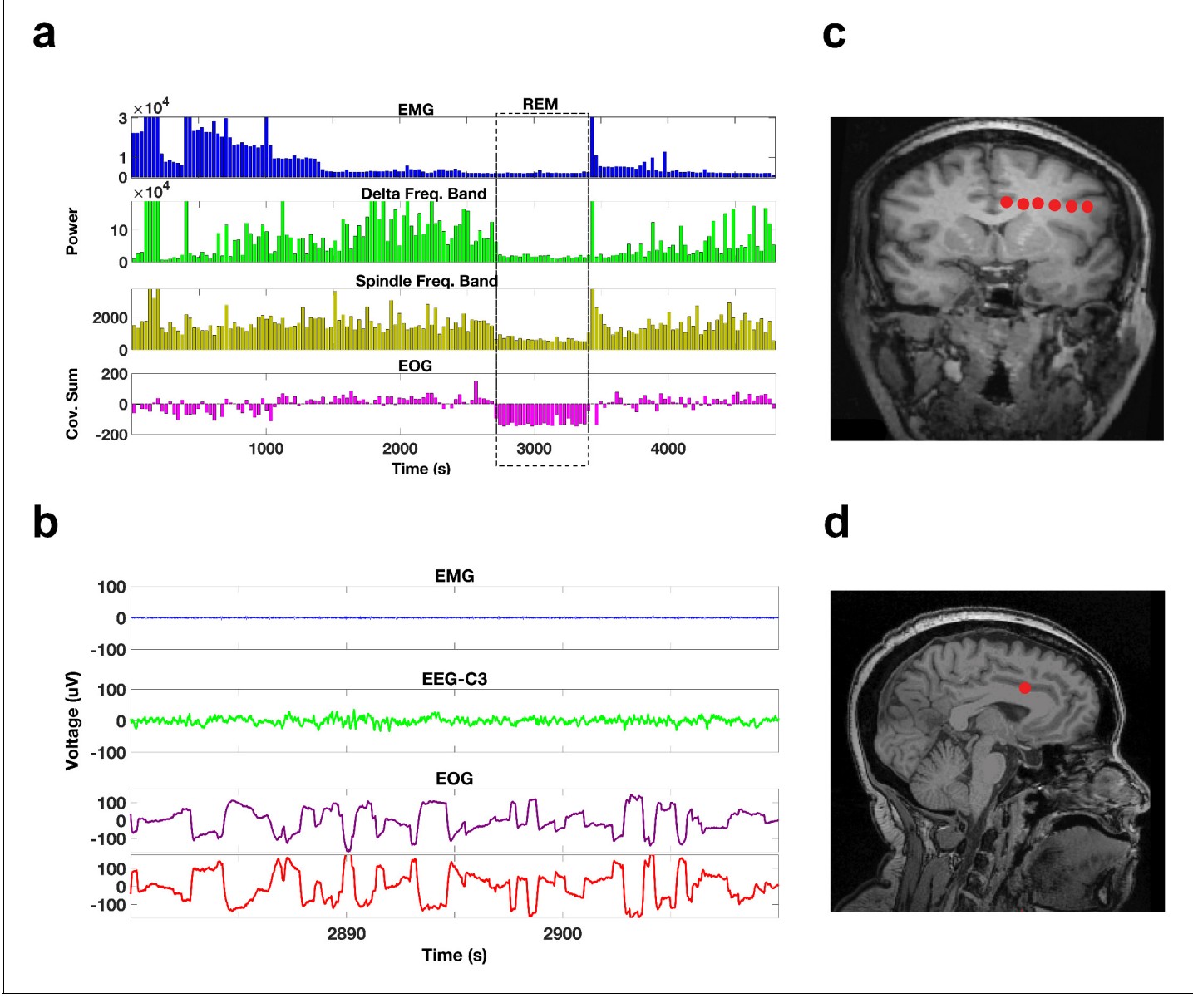

**Figure 1.** Sleep recordings and electrode localization. (**a**) EMG, delta band, and spindle band power, and the cross-covariance of the EOG leads during a portion of a night's sleep from one subject (bin size 30 s). The dotted box indicates a period of REM sleep. During this REM period, EMG, delta, and spindle band powers are all relatively low and the sums of the cross-covariance functions over the 30 s bins are negative. The EOG montages are arranged such that rapid eye movements result in voltage deflections of the opposite polarity resulting in negative cross-covariance values during rapid eye movements. (**b**) Voltage traces of the EMG, scalp EEG, and EOG (purple trace for left eye, red trace for right eye) during a 30 s period from the REM episode demarcated by the dotted box in (**a**). Note that the EMG is of relatively low voltage, the EEG is of low voltage and irregular, and there are rapid eye movements in the EOG with the deflections in the right and left EOG of opposite polarity. (**c**) A coronal MRI image showing the locations of the electrode contacts (red dots) of a frontal electrode in one of the subjects. (**d**) A sagittal MRI image showing the location of the most medial contact in (**c**), which is located in the anterior cingulate.

electrodes showed significant beta activity (median 20.9 Hz, *Figure 3c,d*). The frequency range of the beta activity could be clearly distinguished from the frequency range of the spindling activity that occurs during NREM sleep (*Figure 3—figure supplement 1*).

We then examined electrode contacts in the ACC, which is thought to be relatively active during REM sleep. They also showed significant activity in the beta and theta bands (*Figure 4a,b*). All of the ACC contacts (22/22) showed significant beta activity ($p < 0.05/22$, Bonferroni corrected for

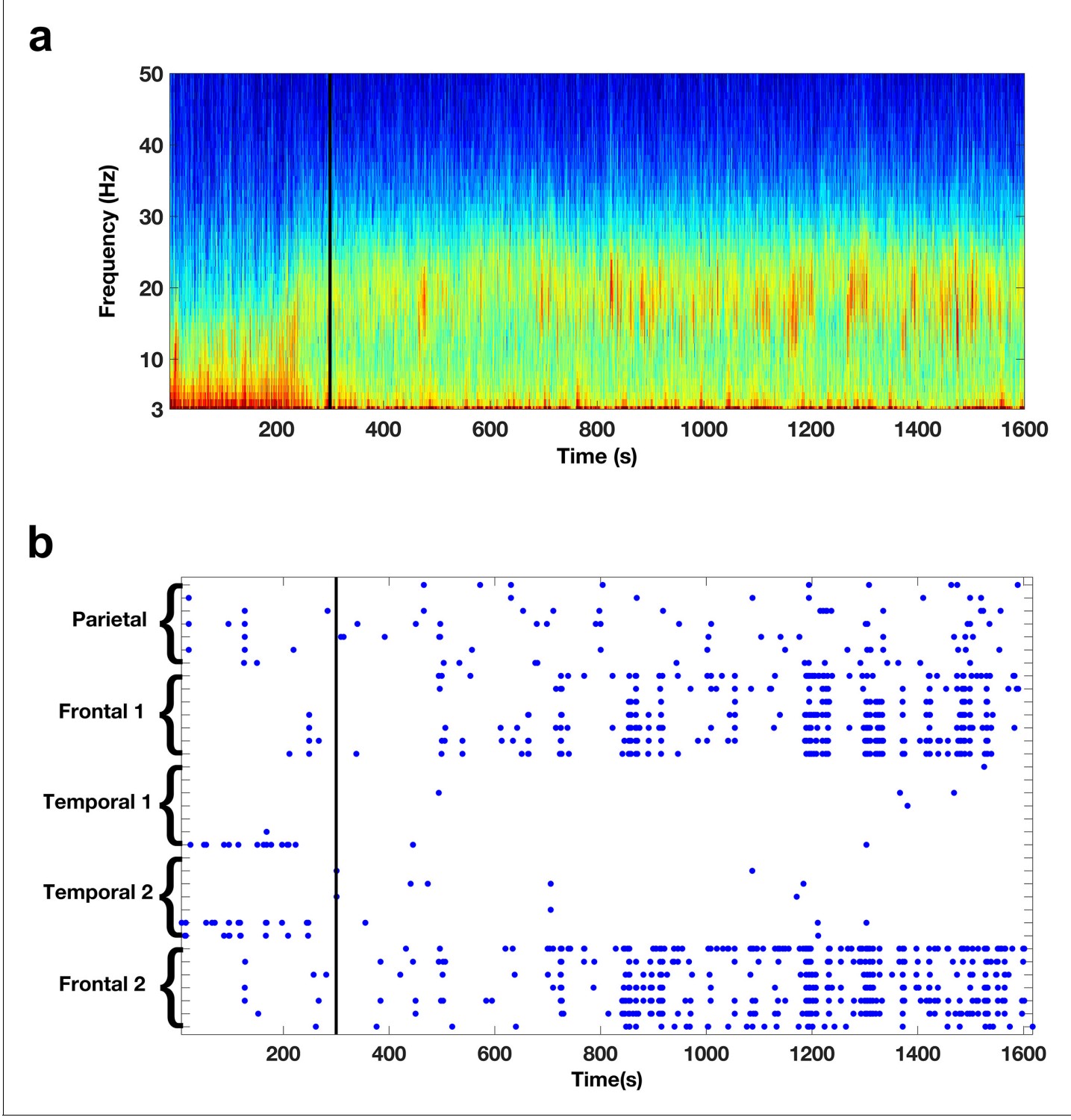

**Figure 2.** Oscillatory activity across scales. (a) Spectrogram shows the activity pattern detected from a bipolar derivation of an electrode located in the frontal cortices during REM sleep. The black line indicates the beginning of the REM sleep episode. (b) Pattern of beta activity across all the electrode contacts of one subject. There are two frontal depth electrodes (Frontal 1 and Frontal 2), two temporal depth electrodes (Temporal 1 and Temporal 2), and one parietal depth electrode (Parietal). Each depth electrode has eight contacts. Each row represents activity from one bipolar derivation. Blue dots indicate periods in which beta power in a given contact was 3.0 standard deviations above the median. The black line indicates the beginning of the REM sleep episode.

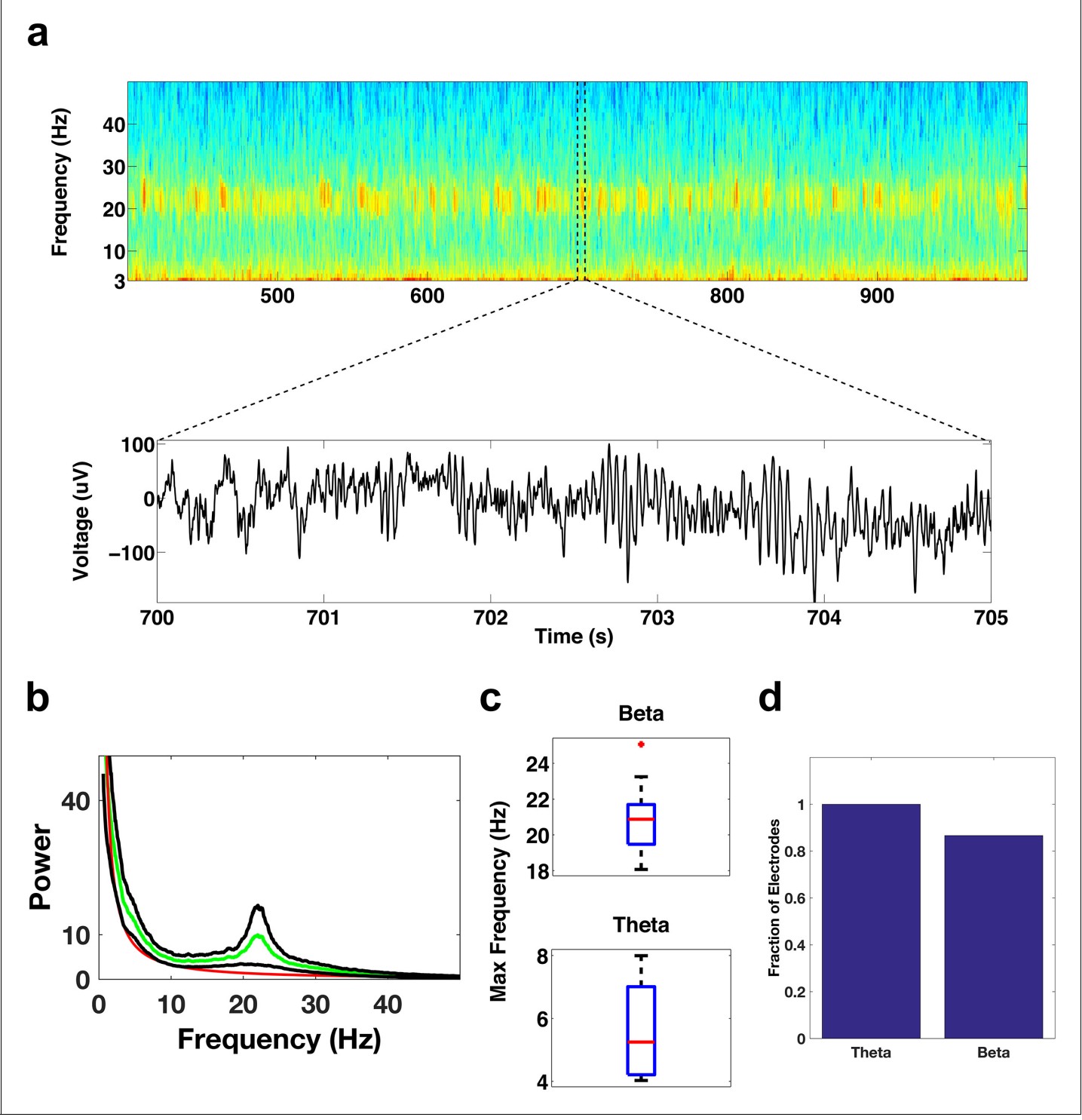

**Figure 3.** Oscillatory activity in the DLPFC. (**a**) Spectrogram shows the activity pattern of an electrode in the DLPFC during a period of REM sleep. The lower panel shows the voltage trace during the period demarcated by the dotted lines in the spectrogram. (**b**) Average power spectrum across all subjects (green trace) of electrodes located in the DLPFC. The red trace is the average of the fits of the power spectra to the model $a*f^b$ and the black traces are ± the standard error of the mean. (**c**) Whisker plot of the peak frequency in the beta band (top) and peak frequency in the theta band (bottom) for electrodes showing significant power in those bands. (**d**) Bar graph showing the fraction of total electrodes in the DLPFC with significant beta or theta power.

The following figure supplements are available for figure 3:

*Figure 3 continued on next page*

*Figure 3 continued*

**Figure supplement 1.** 1/f fit of power spectrum.
**Figure supplement 2.** Comparison of REM and NREM power spectra.

number of electrodes (22), median 19.7 Hz, *Figure 4c,d*) and ~86% (19/22) showed significant theta activity (median 4.6 Hz, *Figure 4c,d*). Significant activity in the beta and theta bands was seen not only in the DLPFC and ACC areas of the frontal cortices, but also in the inferior frontal gyrus (*Figure 4—figure supplement 1*).

We next explored whether theta and beta activity is as prevalent outside the frontal cortices. Theta activity was prevalent in the medial temporal regions, but not beta activity. For example, in the medial temporal gyrus we found ~58% (11/19) of the electrode contacts had prominent theta activity ((p<0.05/19, Bonferroni corrected for number of electrodes (19), median 6.1 Hz, *Figure 5a–c*), but only ~16% (3/19) had prominent beta activity (median 17.9 Hz, *Figure 5a–c*).

We then examined whether the DLPFC and the ACC interacted, since bursts of oscillatory activity appeared to occur in concert between the two areas (*Figure 6a*). We found that a large number (23/31, ~74%) of ACC-DLPFC electrode pairs showed a significant coherence in the beta band (p<0.05, Bonferroni corrected for number of electrode pairs (31 and 34 for beta and theta respectively), *Figure 6b,c*) and 26/34 (~76%) showed a significant coherence in the theta band (*Figure 6b,c*). We found the same results whether we employed the confidence levels provided by Chronux or a Monte Carlo simulation method (see Methods for more details).

We then attempted to determine if either the beta or the theta activity occurs first in either the DLPFC or the ACC, on average. We looked at the cross-covariance in theta power and in beta power between all electrode pairs that showed significant coherence. Both theta and beta cross-covariance functions showed a central peak. The theta peak occurred at approximately 22 ms (*Figure 7a*), suggesting that theta activity on average tended to occur first in the DLPFC. The peak of the beta cross-covariance function occurred very close to zero but was slightly positive (2 ms, *Figure 7b*), suggesting that the beta activity occurred nearly simultaneously in the DLPFC and ACC but that on average it occurred very slightly earlier in the DLPFC. To further examine the difference in the timing of the occurrence of beta and theta activity, we quantified the asymmetry in the cross-covariance function in the −100 to 100 ms window. In particular, we subtracted the area under the cross-covariance curve from −100 to 0 ms from that of 0 to 100 ms. A positive value would suggest that DLPFC activity preceded ACC activity and a negative value the opposite. We found that 69% (18/26, *Figure 7c*) of electrode pairs had a positive value with respect to theta, while 65% (15/23, *Figure 7d*) of electrode pairs had a positive value with respect to beta, suggesting that although on average theta activity and beta activity tended to occur first in the DLPFC, there was a fair amount of variability in timing among the electrodes.

## Discussion

Here we present the novel finding of a theta and beta network in the frontal cortices of humans during REM sleep. The network includes the DLPFC, which prior studies had suggested was relatively quiescent during REM sleep, and the ACC. The bursts of theta and beta activity in the DLPFC and the ACC are coherent between these structures. We suggest that interactions between these two areas may play an important role in the function served by REM sleep.

### DLPFC activity during REM sleep

Imaging studies, in particular Positron Emission Tomography (PET) studies, suggest that the DLPFC is relatively quiescent during REM sleep in comparison with quiet awake periods (*Braun et al., 1997*; *Maquet et al., 1996*; *Muzur et al., 2002*). The relative silence of the DLPFC, a structure thought to be important for executive functions, during REM sleep has been postulated to underlie the often bizarre and illogical nature of our dreams (*Muzur et al., 2002*). Our data suggest that the DLPFC is actually active during REM sleep, since it displays significant theta and beta oscillations (*Figure 3a, b,d*). Furthermore, the DLPFC might be communicating with other structures, since theta and beta

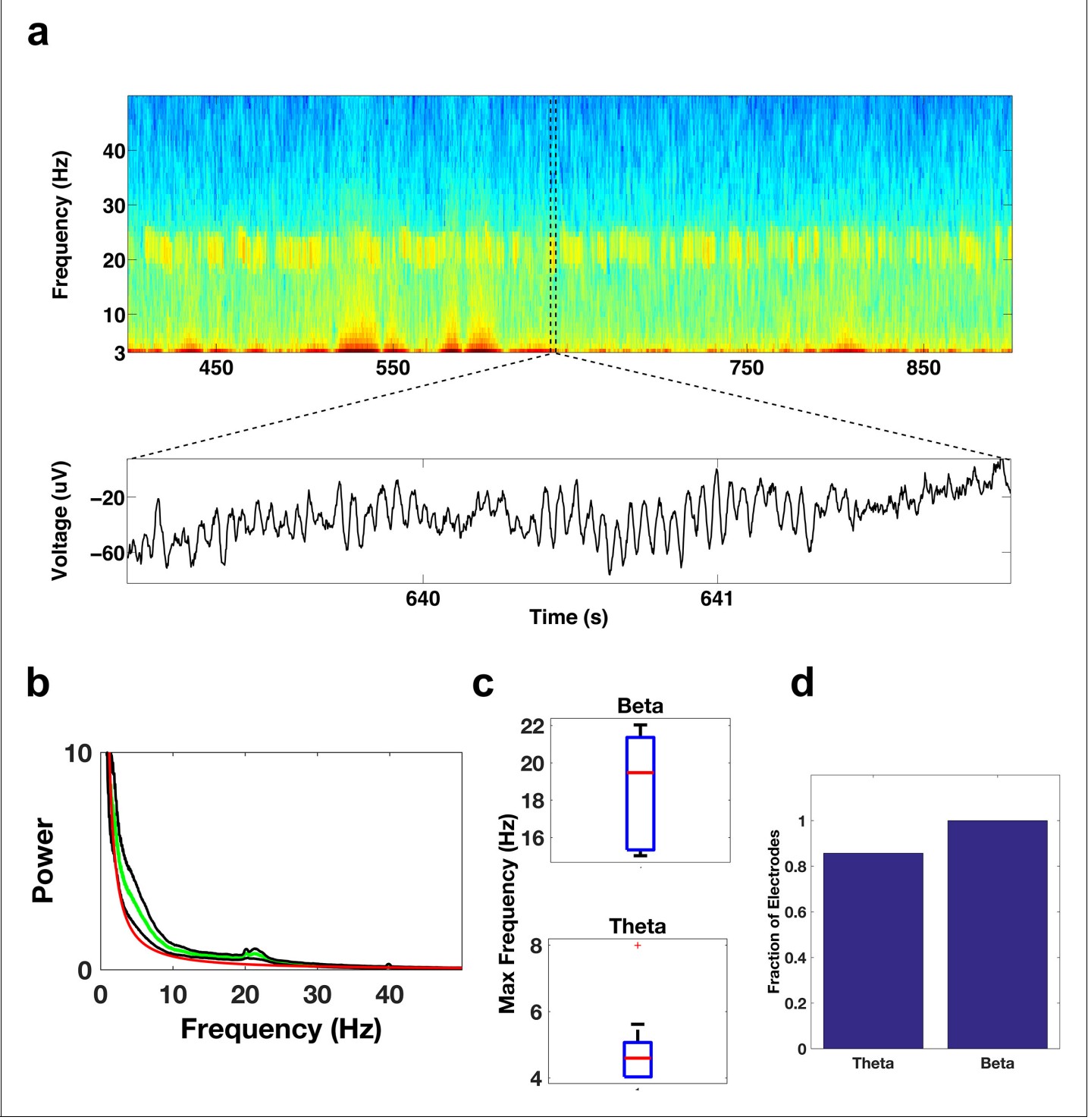

**Figure 4.** Oscillatory activity in the ACC. Same as *Figure 3a–d*, but for electrode contacts in the ACC.

The following figure supplement is available for figure 4:

**Figure supplement 1.** Oscillatory activity in the IFG.

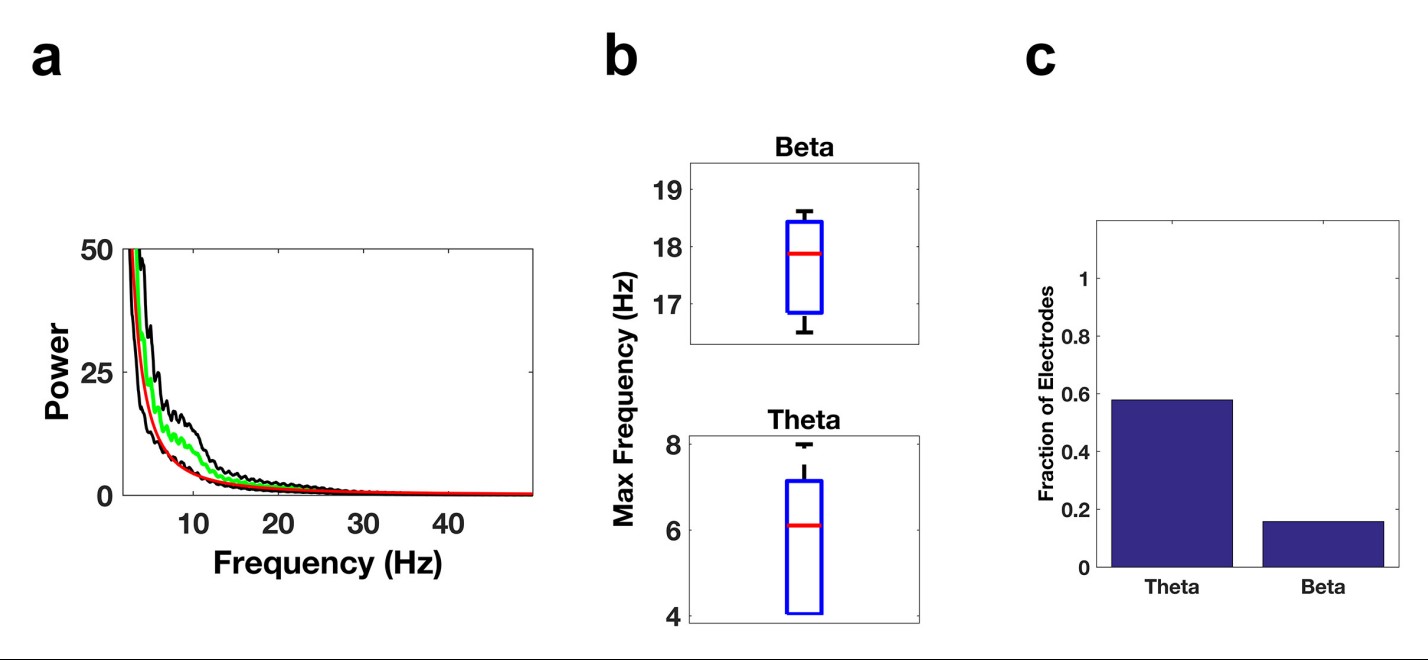

**Figure 5.** Oscillatory Activity Outside the Frontal Cortices. (a–c) Same as *Figure 3b–d*, but for electrode contacts in the middle temporal gyrus.

bands in the DLPFC are coherent with those in the ACC (*Figure 6a–c*). The DLPFC might communicate directly with the ACC or it might be driven by a third structure that coordinates the activity of the DLPFC with other areas. A discrepancy in brain activity levels assessed by PET studies and electrophysiological techniques has also occurred with respect to sleep spindles (*Dang-Vu et al., 2010*). PET studies suggest that the thalamus is relatively quiet during sleep spindles (*Hofle et al., 1997*), while a large body of animal studies shows that spindles are generated precisely by thalamo-cortical interactions (*McCormick and Bal, 1997*; *Steriade et al., 1985*, *1987*). One possibility is that during REM sleep the periods outside the oscillatory events may show less overall neural activity than is seen during awake periods, so that PET measurements, which average over relatively large periods of time (on the order of minutes), indicate that the DLPFC is less active during REM sleep than during quiet awake periods. Another possibility is that the oscillatory activity observed in the DLPFC during REM sleep may manifest under relatively inhibited conditions; the connectivity patterns between the DLPFC and the ACC suggest that the DLPFC may receive more inhibition from the ACC under the physiological conditions of REM sleep in comparison with awake conditions (*Medalla and Barbas, 2012*).

## Functional implications

What functional role sleep plays has yet to be resolved. There is a large body of evidence suggesting that NREM sleep may play an important role in memory consolidation (*Rasch and Born, 2013*), but whether or not REM sleep also plays a role in memory consolidation remains unclear (*Dudai et al., 2015*; *Rasch and Born, 2013*). Some animal and human studies suggest that REM sleep plays a role in procedural and emotional memory consolidation (*Fischer et al., 2002*; *Fu et al., 2007*; *Karni et al., 1994*; *Louie and Wilson, 2001*; *Nishida et al., 2009*; *Nitsche et al., 2010*; *Popa et al., 2010*; *Smith, 2001*; *Wagner et al., 2001*), but other studies, especially human studies, do not corroborate these findings and suggest that REM sleep may be important for other cognitive functions (*Cai et al., 2009*; *Walker et al., 2002b*) such as emotional regulation (*Baran et al., 2012*) while stage 2 NREM sleep may be more important for the consolidation of procedural memories (*Mantua et al., 2016*; *Tamaki et al., 2008*; *Walker et al., 2002a*). Whether these differential findings arise because of the different natures of the tasks, differences across species, or simply because REM sleep plays little or no role in memory consolidation remains to be determined. Below we

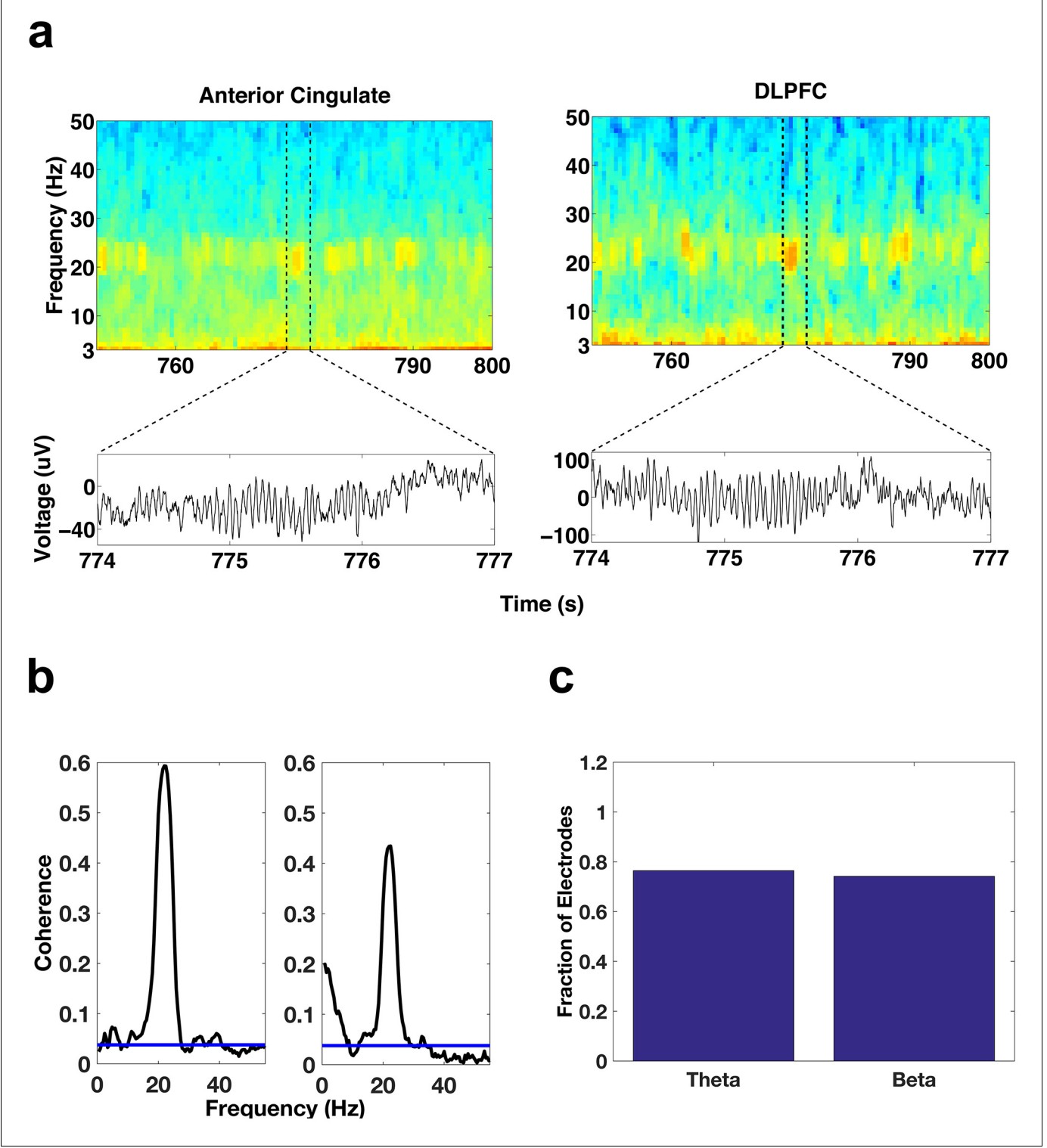

**Figure 6.** Simultaneous oscillatory activity in the DLPFC and ACC. (a) Spectrograms showing the simultaneous activity patterns of an electrode in the ACC (left) and an electrode in the DLPFC (right) during a period of REM sleep. Lower panels show the voltage traces during the periods demarcated by the dotted lines in the spectrograms. (b) Coherence during REM sleep between the DLPFC electrode depicted above and the two ACC contacts on the ipsilateral side of the DLPFC contact. The coherence plot on the left is for the ACC contact depicted above. (c) Bar graph of the fraction of total ACC-DLPFC electrode pairs showing significant theta or beta coherence.

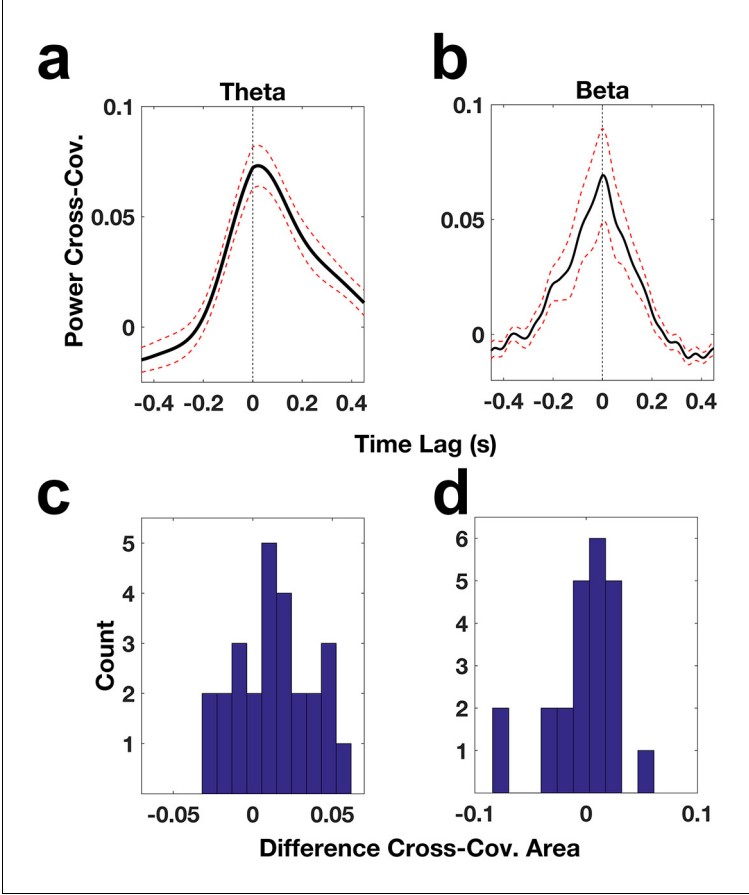

**Figure 7.** ACC and DLPC oscillatory power relationships. (**a**) Cross-covariance of theta power across ACC-DLPFC electrode pairs (solid black trace). The dotted red lines correspond to the standard error of the mean and the dotted black line indicates the zero lag time point. (**b**) Same as (**a**) but for beta power. (**c**) Histogram of the difference between the area under the theta power cross-covariance function from −100 ms to 0 ms and the area under the function from 0 ms to 100 ms. (**d**) Same as (**c**) but for the beta power cross-covariance function.

discuss our findings in light of previously known facts about the structures and oscillations at play during REM sleep. Our discussion is driven by the perspective that REM oscillatory activity may play an important role in memory, and we put forward hypotheses for how the rhythmic activity that we have observed during REM sleep could serve a role in memory consolidation, but we note that rhythmic activity during REM sleep could serve some other function or possibly no function at all.

We observed rhythmic activity in the beta and theta bands that was coherent between the ACC and the DLPFC, two relatively distant structures that are reciprocally connected and both implicated in memory (*Goldman-Rakic, 1995*). It is possible that this activity could be important in the procedural motor memory consolidation since the ACC is interconnected with motor areas (*Bates and Goldman-Rakic, 1993*; *Dum and Strick, 2002*, *1991*; *Hatanaka et al., 2003*; *Morecraft and Van Hoesen, 1993*, *1992*; *Picard and Strick, 1996*), is thought to have motor regions itself (*Paus, 2001*), and is thought to be important in error detection (*Holroyd and Yeung, 2011*; *Paus, 2001*). In humans, beta activity has been shown to follow PGO waves during REM sleep in the subthalamic nucleus of the basal ganglia, so the beta activity observed in our work could result from the transmission of the PGO-related beta activity from the basal ganglia (*Amzica and Steriade, 1996*; *Fernández-Mendoza et al., 2009*); pontine wave density has been correlated with sleep-related improvements in memory tasks and the expression of plasticity-related genes (*Datta, 2000*; *Datta et al., 2008*). This possibility is in line with our findings that the observed beta activity tends to manifest at approximately the same time in both the ACC and the DLPFC, suggesting that it may

originate from a subcortical source. Such a scenario leads to the intriguing possibility that PGO waves may trigger an evaluation of a motor plan by a dialogue occurring via the beta band channel (*Figure 8*).

With respect to theta, experimental studies suggest that the hippocampus provides contextual and spatial information to the cingulate and prefrontal cortex via the theta band (*Jones and Wilson, 2005*; *Remondes and Wilson, 2013*) and disrupting hippocampal theta activity during REM sleep results in contextual memory deficits (*Boyce et al., 2016*). Therefore, the theta band may be a channel through which spatial content, e.g., the location at which a procedural motor task was acquired, is provided to the ACC.

With respect to emotional memories the theta band may be a channel through which the amygdala provides the emotional content or valence of a particular memory to the ACC since both human and animal studies suggest that prefrontal theta and the interaction of prefrontal theta activity with both the hippocampus and the amygdala may be important for the consolidation of emotional memories during REM sleep (*Fu et al., 2007*; *Nishida et al., 2009*; *Popa et al., 2010*; *Rasch and Born, 2013*; *Wagner et al., 2001*). Furthermore, emotional information from the limbic system conveyed via the theta band may bias the processing and consolidation of procedural memories with high emotional content (*Popa et al., 2010*). Note that many different functions have been attributed to

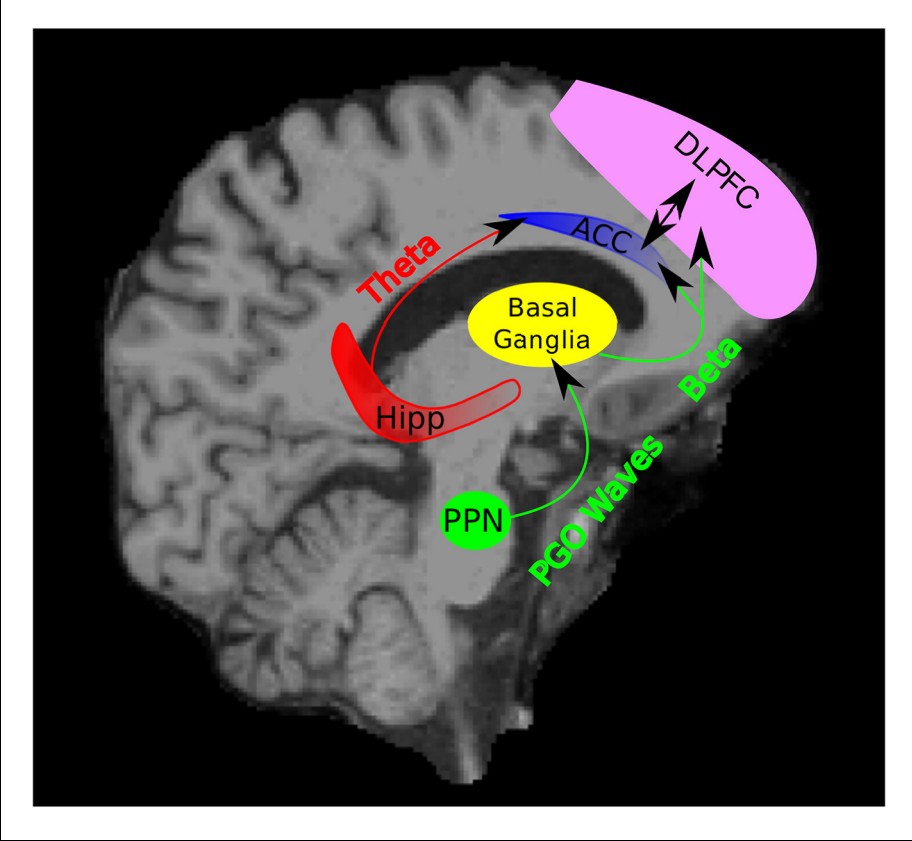

**Figure 8.** Schematic of Hypothesized Circuit Involved in REM Sleep Memory Consolidation. PGO waves originating from the PPN may trigger beta activity in the basal ganglia. This beta activity in the basal ganglia may then be transmitted to cortical areas including the ACC and DLPFC. The triggering of the beta activity by the PGO waves may begin a dialogue between the basal ganglia, ACC, and DLPC (as well as other areas) through the beta band in which a motor plan is evaluated, ultimately leading to motor memory consolidation. Theta activity may be a channel through which the hippocampus provides contextual and spatial information about a given memory to the ACC and other parts of the frontal cortex. (PPN = Pedunculopontine tegmental nucleus and Hipp = Hippocampus).

the ACC (*Paus, 2001*), so its activation during REM sleep does not necessarily mean that it serves the function of emotional or procedural memory consolidation during REM sleep.

## Conclusion

We generally lack detailed knowledge about the physiological characteristics of REM sleep, especially in humans. Recent intracranial studies in humans have begun to provide us with more detailed information about the physiological processes underlying REM sleep (*Andrillon et al., 2015*; *Cantero et al., 2003*). These studies have primarily focused on REM-related processes (both at the unit and LFP level) in the medial temporal lobe. We build on this knowledge base by examining the spatio-temporal characteristics of rhythmic activity in the frontal cortices. Here we have made significant inroads toward better understanding the physiological processes underlying REM sleep through the discovery of a frontal beta-theta network and we hypothesize that this network might mediate memory consolidation during REM sleep.

# Materials and methods

## Subjects and recordings

Data were collected from five patients (four females and one male) between the ages of 42 and 55 years (mean 48.0 ± 5.6 (SD) years) undergoing invasive monitoring using intracranial depth electrodes for intractable epilepsy (see *Table 1*). Each depth electrode had 6 to 8 contacts. Subject 1 had 10 depth electrodes (two anterior temporal (bilateral), two posterior temporal (bilateral), and six frontal [bilateral]), subject 2 had eight depth electrodes (two anterior temporal (bilateral), two posterior temporal (bilateral), four frontal [bilateral]), subject 3 had five depth electrodes (one anterior temporal, one posterior temporal, two frontal, and one parietal), subject 4 had 10 depth electrodes (two anterior temporal (bilateral), two posterior temporal (bilateral), and six frontal [bilateral]), and subject 5 had 10 depth electrodes (two anterior temporal (bilateral), two posterior temporal (bilateral), and six frontal [bilateral]). The placement of the electrodes was determined strictly by clinical criteria. The patients gave informed consent. The research protocol was approved by the Partners Human Research Committee.

The contacts on the depth electrodes were spaced by 10 mm and had an impedance of ~100 Ω. The electrophysiological signals were sampled at 500 Hz (four subjects) or at 512 Hz (one subject).

## Electrode localization

The locations of the depth electrode contacts were determined using a preoperative high-resolution T1-weighted magnetic resonance imaging (MRI) scan and a postoperative computerized tomography (CT) scan. The preoperative MRI scan was used to generate a 3D rendering of the brain using FreeSurfer (*Dale et al., 1999*; *Fischl et al., 1999*); this rendering was placed in a 3D coordinate system (RAS). The postoperative CT scan was then coregistered with the 3D rendering, thus providing a RAS coordinate for each electrode (*Dykstra et al., 2012*). FreeSurfer automatically generated sub-cortical and cortical parcellations with anatomical labels using the Desikan-Killiany Atlas (*Desikan et al., 2006*; *Fischl et al., 2002*). Using these parcellations and the RAS coordinates, we determined the anatomical location of each electrode contact. We considered contacts labeled as being in rostral middle frontal by FreeSurfer according to the Desikan-Killiany Atlas (*Desikan et al., 2006*) to be in the DLPFC. We considered the inferior frontal gyrus to consist of contacts labeled as parstriangularis, parsopercularis, and parsorbitalis. Contacts labeled as middle temporal gyrus were categorized as such. The locations of the contacts were also determined by a trained neurologist. In the case of the midline contacts (in relation to this paper, the anterior cingulate contacts) there was some discrepancy between the locations of the contacts determined by the neurologist and by the FreeSurfer labeling scheme. The locations determined by the neurologist were used. We employed bipolar derivations using two abutting contacts so that we could be sure the activity being analyzed was relatively local in nature. Only electrode derivations for which both contacts had labels within the same defined region and such that at least one of the contacts was in the gray matter were used for subsequent analyses. We did not use bipolar derivations with contacts in two different regions since in these cases it was not possible to determine from which region the activity originated. In addition, contacts that showed epileptic activity were eliminated from subsequent analyses (36

bipolar derivations were eliminated by this criterion). After eliminating bipolar derivations based on these criteria, we were left with 15 bipolar derivations in the DLPFC, 22 in the ACC, 12 in the inferior frontal gyrus, and 19 in the medial temporal gyrus. In other defined regions, there were relatively few bipolar derivations that met the criteria for inclusion in our analyses, and not all subjects had bipolar derivations in these regions.

## Sleep staging

Scalp electroencephalography (EEG), electromyography (EMG), and electrooculography (EOG) were used to score sleep. Sleep was first scored using custom written software (https://github.com/svi-jayan9/Sleep-Viewer) that incorporated the criteria specified by the American Association of Sleep Medicine for sleep scoring (*Iber et al., 2007*). Those periods specified as REM sleep were subsequently examined by eye and verified as REM sleep epochs by experienced sleep scorers (SSC). We also verified via video recordings that the patients were asleep during identified REM sleep epochs. The subjects spent 73.18% ± 6.82% (SEM) of the time in NREM sleep, 16.23% ± 2.85% of the time in REM sleep, and 10.59% ± 1.78% of the time awake after sleep onset (WASO), with a total sleep time of 378.13 ± 22.67 min. We used conservative measures in identifying REM periods for the purposes of further analysis. We chose relatively long (>5 min) REM periods that were not interrupted by non-REM sleep stages. Furthermore, we selected the beginning and the end of each REM period (first and last 10 s) such that there were no other stages or transition stages present. Thus, our results were not influenced by the spectral content from any other sleep stages.

## Data analysis

Those electrode contacts identified to be in the epileptic focus or identified as having interictal activity by the epileptologists were excluded from further analysis. In addition, nights during which seizures occurred were excluded from further analysis. All analyses were done using a bipolar montage, specifically by subtracting adjacent contacts.

## Identification of contacts with significant theta and beta power

REM episodes were divided into 10 s windows. For each electrode, a power spectrum estimate, $S_{e,j}(f)$, was calculated for each of the ten-second windows using the Chronux (*Bokil et al., 2010*) toolbox for Matlab (MathWorks). Here, $e$ is the electrode number ($e \in \{1, \ldots, n\}$) and $j$ specifies the trial number ($j \in \{1, \ldots, J\}$). For each 10 s window, the power spectrum estimate was fit as a function of frequency, $f$, to the model $af^b$, using a robust least-squares regression to obtain $S_{e,j}^r(f)$, the robust fit for a given electrode and trial (see *Figure 3—figure supplement 1*). For each frequency band of interest (e.g., $f \in [4, 8]$ for theta), the mean values $\bar{S}_{e,j}$ and $\bar{S}_{e,j}^r$ were calculated for each 10 s window for both the spectrum estimates $S_{e,j}(f)$ and the fitted curves $S_{e,j}^r(f)$. For each electrode, a t-test was used to determine whether the set of mean values of the spectrum estimates for a given band was significantly different (p<0.05, Bonferroni corrected for number of electrodes) from the set of mean values of the fitted curves. We used this method rather than a comparison with awake activity since during awake conditions the calculated power spectrum would depend on the cognitive state and the task being performed. See *Figure 3—figure supplement 2* for power spectrum comparisons between NREM sleep and REM sleep, which demonstrate that the spectral content is very different between the two states and show the clear distinction between spindling activity during NREM sleep and beta activity during REM sleep. Note that all subjects had at least 21 min of REM sleep, resulting in over 125 10 s windows per electrode in each subject. Only electrode contacts showing significant power in the band of interest were used for subsequent analysis. However, this resulted in the elimination of very few electrodes. In the DLPFC no electrodes were eliminated from the theta band analyses (0/15) and two electrodes were eliminated from the beta band analyses (2/15). In the ACC no electrodes were eliminated from the beta band analyses (0/22) and three electrodes were eliminated from the theta band analyses (3/22).

## Coherence estimates

We estimated coherence values using the Chronux toolbox. Coherence was examined for 34 ACC-DLPFC electrode pairs in the theta band (3 ACC electrodes were eliminated because theta power was not significant) and 31 ACC-DLPFC electrode pairs in the beta band (2 DLPFC electrodes were

eliminated because beta power was not significant). We estimated the coherence during REM sleep by first dividing all REM sessions of a subject into 2 s windows. All 2 s intervals of data were included in a single coherence calculation for that subject using the coherence formula employed by Chronux (see equations 7.77 and 7.80 in *Mitra and Bokil, 2008*). The time-bandwidth was set such that the coherence estimate for the center band (e.g., 6 Hz for theta) integrated the entire band of interest. We assessed significance using two methods: by using the confidence level provided by Chronux and by doing Monte Carlo simulations (2000 shuffles). The confidence level provided by Chronux is based on the distribution of the statistic (i.e., the coherence measure) given the null hypothesis of zero coherence (*Bokil et al., 2010*; *Brillinger, 2001*; *Jarvis and Mitra, 2001*). The distribution of the estimator is given by the following formula.

$$p\left(|C|^2\right) = \frac{1}{m-1}\left(1 - |C|^2\right)^{m-2}$$

In the formula m is the degrees of freedom, which is equal to the number of 2 s intervals multiplied by the number of data tapers used in the calculation. See *Brillinger (2001)* for details regarding the asymptotic convergence of the coherence estimator under the hypothesis of zero coherence. For the Monte Carlo simulations, a shuffle was done by permuting the 2 s windows of one of the electrode pairs and then calculating the coherence. Using the distribution of coherences from the shuffles we determined the probability of getting the coherence value we got using our data. Both methods identified the same set of DLPFC-ACC electrode derivation pairs as having significant (p<0.05, Bonferroni corrected for number of electrode pairs) beta or theta coherence.

## Power Cross-covariance estimates

The cross-covariance of the power was calculated by first filtering the raw signal in the band of interest. The instantaneous amplitude was then calculated using the Hilbert transform of the filtered signal. Using these calculated instantaneous amplitudes the cross-covariance of the power (amplitude) of an ACC-DLPFC electrode pair was calculated for each two-second REM sleep window. For a given electrode the cross-covariance functions for all two-second windows were averaged to obtain an average cross-covariance function.

Only those REM windows in which both electrodes had power in the top quartile in the band of interest with respect to the REM episode were used for the analysis. We also restricted our analysis to those electrode pairs that showed a significant coherence in the band of interest.

Finally the average cross-covariance functions for all electrodes were averaged to obtain an overall average cross-covariance function. The max lag reported is simply the time lag at the maximum value of the overall average cross-covariance function. We used 23 ACC-DLPFC bipolar electrode pair derivations for the beta band cross-covariance analysis and 26 ACC-DLPFC bipolar electrode pair derivations for the theta band cross-covariance analysis (we omitted eight electrode pairs from the theta and beta analyses).

## Acknowledgements

SV, KL, and NK acknowledge support from the National Science Foundation (NSF) Grant DMS-1042134. SSC received support for this work from ONR N00014-13-1-0672 and an MGH ECOR Research Scholars award. The authors thank Anna Vijayan, Giovanni Piantoni, Jung H Lee, Benjamin Pittman-Polletta, Michelle M McCarthy, Jean-Baptiste Eichenlaub, Salva Ardid, Jason Sherfey, and David Stanley for their helpful comments on this work. The authors would also like to thank Brandon Lockyer for helpful input on sleep staging.

## Additional information

### Funding

| Funder | Grant reference number | Author |
|---|---|---|
| National Science Foundation | DMS-1042134 | Sujith Vijayan<br>Kyle Q Lepage<br>Nancy J Kopell |

| Office of Naval Research | N00014-13-1-0672 | Sydney S Cash |

The funders had no role in study design, data collection and interpretation, or the decision to submit the work for publication.

## Author contributions

SV, Conceptualization, Formal analysis, Investigation, Methodology, Writing—original draft, Project administration, Writing—review and editing; KQL, Formal analysis; NJK, SSC, Funding acquisition, Writing—review and editing

## Author ORCIDs

Sujith Vijayan, http://orcid.org/0000-0001-6385-3922

## Ethics

Human subjects: All patients gave informed consent. The research protocol was approved by the Partners Human Research Committee. IRB number 2007P000165.

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
