## [Decision Letter]

Thank you for submitting your article "Frontal Βeta-Theta Network during REM Sleep" for consideration by *eLife*. Your article has been reviewed by three peer reviewers, one of whom is a member of our Board of Reviewing Editors, and the evaluation has been overseen by Sabine Kastner as the Senior Editor.

The reviewers have discussed the reviews with one another and the Reviewing Editor has drafted this decision to help you prepare a revised submission.

Summary:

This manuscript presents a study in which intracranial recordings of REM sleep were conducted in humans (epileptic patients). Based on neuroimaging studies, the authors have thought of the ACC and DLPFC as regions of specific interest, which may be inversely active to each other (ACC high, DLPFC low) during REM. However, the results of the present study suggest that both areas are active, with significant beta and theta activity that occurs coherently between the regions. The authors suggest that the results indicate a neurological basis for a proposed function of sleep in memory consolidation.

All reviewers acknowledge the importance of spatio-temporal analyses of spontaneous oscillations during REM sleep in human. However, a multitude of analyses and statistical procedures are unclear.

Essential revisions:

1) All reviewers have agreed that significantly better clarification of analyses and control analyses are necessary.

For example:

“With regards to the data analysis, it's not clear how conclusions such as "showed significant theta activity" are drawn. It seems that frequency analyses were done in either 10- or 30-sec bins. Either way, a bin may not be 100% REM. Presumably the 'edges' on the sleep stage would have some other sleep (and frequencies). And, what make it 'significant'? A t-test is used (as indicated by the reported statistic) but this is not in the Data Analysis section”.

“It was not clear how many out of all channels were analyzed and where these channels were placed (scalp locations). "Frontal cortices" usually indicates a huge extent and thus ambiguous. How specific could the ACC and DLPFC be in comparison to other frontal regions (superior frontal cortex, orbitofrontal cortex etc.)? Is it the case that the ACC and DLPFC were more active compared to other frontal cortices? Or is it the case that the ACC and DLPFC were as active as other frontal cortices but more active than other non-frontal cortices? It is stated that Bonferroni correction was made based on the number of electrodes. However, the number of electrodes is not reported”.

“The authors limited their analysis to a potentially quite restricted subsample of the data. First, data from frontal electrodes were reduced to electrodes in the ACC and DLPFC, because beta and theta activity was most prominent in these regions. From these electrodes, only those were used for coherence analysis that showed a significant power increase in the frequency bands of interest. From these electrode pairs, only those that showed significant coherence were used for the subsequent power cross-covariance analysis, which was moreover only calculated for time windows with power in the top 25% within each REM episode. All of this data-driven sub-selection potentially bias the results towards positive findings. As an example, using only electrode pairs with high coherence for power cross-covariance can be problematic since these two measures are not independent (Lachaux, Rodriguez et al. 1999, Srinath and Ray 2014). Furthermore, it is not clear how much data (sleep epochs, electrodes, time windows, electrode pairs) were discarded in absolute numbers in each step (only the ratios of electrodes used based on power and coherence are given in Figure 3–Figure 6). Please report the number of recorded electrodes per subject including their location. Please give an idea on how many data were finally used at each analysis step. Coherence was tested for statistically significant differences using Monte Carlo simulations as well as the "confidence level provided by Chronux" (subsection “Coherence Estimates”). What is this confidence level based on? Apparently, both methods yielded "the same results" (Paragraph six Results section). Please elaborate, are the results indeed fully identical? How did the authors deal with the influence of the number of trials for the estimation of coherence (e.g. Maris, Schoffelen, Fries 2007). Was the number of observations equalized across REM sleep periods? In the subsection “Coherence Estimates”, the calculation of coherence reads as if the coherence between two electrodes was calculated for each window and then was averaged. However, since calculating coherence for a single trial (here: 2-s time window) will always result in a coherence of 1 (as well as their average), I assume that the cross-spectra were calculated for each time window and then used to calculate coherence (not "average coherence") across time windows. Please clarify the procedure”.

Please state in significantly clearer way how you conducted the analyses, what was compared, or how the statistical significance was obtained. Bin length, definition of regions, data selection (if any), procedures of a coherence analysis, cross-covariance analysis, and data visualization (Figure 2–Figure 6 each show spectral data from only one bipolar derivation or electrode pair. How representative are these data? Is there a way to present data aggregated from more than one electrode / subject?), etc. are unclear. In addition, please clarify whether you compared REM sleep vs. NREM sleep or Wake stage.

2) All reviewers agreed that discussion about REM sleep and memory consolidation are too far-reaching based on the present data, particularly because it is not shown that the obtained effects are specific to REM sleep.

“The function of REM in procedural memory consolidation is questionable as the result is not found in many studies. Recent studies seem to suggest that REMs role in emotional memory consolidation is limited (e.g., see Payne et al., 2015) and perhaps reflects emotional processing without consolidation (see Baran et al., 2012). Keeping in mind that the present data contain no cognitive measures, it cannot be stated "These findings, in conjunction with previous work on REM sleep, suggest a system for the consolidation of memories during REM sleep." It seems that much of the discussion is focused on just this point – the REM role in consolidation – which I see no clear support for in the present data. Sure they support some cognitive role of REM but is it necessarily emotional or procedural memory consolidation? That link is absent”.

“The authors speculate that activation of the ACC during REM sleep relates to consolidation motor learning. However, there are many functions in which the ACC is suggested to play roles. Mere activation of the ACC during REM sleep may leave other possibilities than its involvement in consolidation of motor learning”.

Please revise the relevant statements so that the manuscript would not go too far beyond the current data.

---

## [Author Response]

*Essential revisions:*

*1) All reviewers have agreed that significantly better clarification of analyses and control analyses are necessary.*

We have substantially modified the Methods and Results sections to make our approaches to analyses clearer. We have added much more detail on electrode selection, electrode localization, statistical measures for frequency, coherence and cross-covariance estimates, sleep staging, determination of statistical significance, and more. The changes to the manuscript are detailed below in response to specific reviewer queries.

*For example:*

*“With regards to the data analysis, it's not clear how conclusions such as "showed significant theta activity" are drawn. It seems that frequency analyses were done in either 10- or 30-sec bins. Either way, a bin may not be 100% REM. Presumably the 'edges' on the sleep stage would have some other sleep (and frequencies). And, what make it 'significant'? A t-test is used (as indicated by the reported statistic) but this is not in the Data Analysis section”.*

As stated in the section entitled Determination of Significant Theta and Β Activity, power spectra were calculated in 10-second bins. For each calculated power spectrum a 1/f fit was determined using a robust least-squares regression (see Figure 3—figure supplement 1). We then calculated the mean power for each power spectrum and each fit in the band of interest. We then used a t-test to determine if the set of means of the power spectra differed significantly from the set of means of the fits in the bands of interest. An electrode is said to show significant theta activity if the means of the power spectra in the theta band were significantly different from the means of the fits in the theta band as assessed by a t-test (P<0.05, Bonferroni corrected for number of electrodes).

We used conservative measures in identifying REM periods. We chose relatively long REM periods that were not interrupted by non-REM sleep stages. Furthermore, we selected the beginning and the end of each REM period (first and last 10 seconds) such that there were no other stages or transition stages present. Thus our results were not influenced by spectral content from any other sleep stages.

To address the reviewer's concerns, we have added the relevant details to the section of the Methods entitled Contacts with Significant Theta and Β Power (see the section entitled Determination of Significant Theta and Β Activity for details). To further address the reviewer’s concerns we have added details to the section of the Methods entitled Sleep Staging. The relevant text is pasted below.

*“*We used conservative measures in identifying REM periods for the purposes of further analysis. We chose relatively long (>5 minutes) REM periods that were not interrupted by non-REM sleep stages. Furthermore, we selected the beginning and the end of each REM period (first and last 10 seconds) such that there were no other stages or transition stages present. Thus our results were not influenced by spectral content from any other sleep stages.”

*“It was not clear how many out of all channels were analyzed and where these channels were placed (scalp locations). "Frontal cortices" usually indicates a huge extent and thus ambiguous. How specific could the ACC and DLPFC be in comparison to other frontal regions (superior frontal cortex, orbitofrontal cortex etc.)? Is it the case that the ACC and DLPFC were more active compared to other frontal cortices? Or is it the case that the ACC and DLPFC were as active as other frontal cortices but more active than other non-frontal cortices? It is stated that Bonferroni correction was made based on the number of electrodes. However, the number of electrodes is not reported”.*

Identification of Electrode Localization

Subjects were implanted with depth electrodes. Each depth electrode had 6 to 8 contacts. Subject 1 had 10 depth electrodes (2 anterior temporal (bilateral), 2 posterior temporal (bilateral), and 6 frontal (bilateral)), subject 2 had 8 depth electrodes (2 anterior temporal (bilateral), 2 posterior temporal (bilateral), 4 frontal (bilateral)), subject 3 had 5 depth electrodes (1 anterior temporal, 1 posterior temporal, 2 frontal, and 1 parietal), subject 4 had 10 depth electrodes (2 anterior temporal (bilateral), 2 posterior temporal (bilateral), and 6 frontal (bilateral)), subject 5 had 10 depth electrodes (2 anterior temporal (bilateral), 2 posterior temporal (bilateral), and 6 frontal (bilateral)).

Electrode locations were determined by co-registering the postoperative CT scan with the preoperative MRI scan (see Figure 1 in the manuscript for an example) using techniques previously developed in the lab (Dykstra et al., 2012). In brief, FreeSurfer (Dale et al., 1999; Fischl et al., 1999) was used to create a 3-D rendering of the brain using the MRI scan and place the rendering in a 3-D coordinate system, thus providing a 3-D coordinate for each electrode contact. FreeSurfer was also used to automatically parcellate the cortical and subcortical structures and provide anatomical labels using the Deskian-Killany atlas; from Figure 1 in Deskian et al. (2006). The 3-D coordinate assigned to each electrode contact was used to assign a unique anatomical label to each contact.

We employed bipolar derivations using two adjacent contacts so that we could be sure the activity being analyzed was relatively local in nature. Only electrode derivations for which both contacts had labels within the same defined region and such that at least one of the contacts was in the gray matter were used for subsequent analyses. We did not use bipolar derivations with contacts in two different regions since in these cases it was not possible to determine from which region the activity originated. In addition, contacts that showed epileptic activity were eliminated from analyses (36 bipolar derivations were eliminated by this criterion). After eliminating bipolar derivations based on these criteria, we were left with 15 bipolar derivations in the DLPFC, 22 in the ACC, 12 in the inferior frontal gyrus, and 19 in the medial temporal gyrus. In other defined regions there were relatively few bipolar derivations that met the criteria for inclusion in our analyses; furthermore, not all of the subjects had bipolar derivations in these regions. For instance, other frontal areas (e.g. orbitofrontal cortex) had very few bipolar electrode derivations that satisfied our criteria and each of these areas had electrodes in only 2 or fewer out of 5 subjects.

In this study, scalp electrodes were used only for sleep staging. All other analyses utilized depth electrodes, which do permit anatomical localization using the methods outlined above.

To address the reviewer's concerns, a paragraph has been added to the methods section entitled Subjects and Recordings. The relevant text is pasted below.

“Each depth electrode had 6 to 8 contacts. Subject 1 had 10 depth electrodes (2 anterior temporal (bilateral), 2 posterior temporal (bilateral), and 6 frontal (bilateral)), subject 2 had 8 depth electrodes (2 anterior temporal (bilateral), 2 posterior temporal (bilateral), 4 frontal (bilateral)), subject 3 had 5 depth electrodes (1 anterior temporal, 1 posterior temporal, 2 frontal, and 1 parietal), subject 4 had 10 depth electrodes (2 anterior temporal (bilateral), 2 posterior temporal (bilateral), and 6 frontal (bilateral)), and subject 5 had 10 depth electrodes (2 anterior temporal (bilateral), 2 posterior temporal (bilateral), and 6 frontal (bilateral)).”

To address the reviewer's concerns, a paragraph has also been added to the methods section entitled Electrode Localization. The relevant text is pasted below.

“We employed bipolar derivations using two abutting contacts so that we could be sure the activity being analyzed was relatively local in nature. Only electrode derivations for which both contacts had labels within the same defined region and such that at least one of the contacts was in the gray matter were used for subsequent analyses. We did not use bipolar derivations with contacts in two different regions since in these cases it was not possible to determine from which region the activity originated. In addition, contacts that showed epileptic activity were eliminated from subsequent analyses (36 bipolar derivations were eliminated by this criterion). After eliminating bipolar derivations based on these criteria, we were left with 15 bipolar derivations in the DLPFC, 22 in the ACC, 12 in the inferior frontal gyrus, and 19 in the medial temporal gyrus. In other defined regions there were relatively few bipolar derivations that met the criteria for inclusion in our analyses, and not all subjects had bipolar derivations in these regions.”

Determination of Significant Theta and Beta Activity

For each region of interest (e.g., the DLPFC) all bipolar electrode derivations in the region were tested for significant theta and beta activity. Identified REM periods were divided into 10-second windows (i.e., the bin size was 10s). The power spectrum was calculated for each 10-second window (Figure 3—figure supplement 1) and a 1/f fit was calculated for each calculated power spectrum using a robust least-squares regression (Figure 3—figure supplement 1). To determine if there was significant power in a given band (e.g., theta) we took the mean power within the given band for each spectrum estimate and each fit, for each 10 -second bin. This gave a set of paired values corresponding to the mean powers (for the given band) for the spectrum estimates and for the fitted curves of the various 10- second bins. For each bipolar derivation, a t-test (P<.05, Bonferroni corrected for number of bipolar derivations) was used to determine whether there was significant power in the band of interest.

We used this method rather than a comparison with awake activity since during awake conditions the calculated power spectrum would depend on the cognitive state and the task being performed. We have included in the figure supplements power spectrum comparisons between NREM sleep and REM sleep (Figure 3—figure supplement 2) to demonstrate that the spectral content is very different between the two states and to show the clear distinction between spindling activity (11-16 Hz) during NREM sleep and beta activity during REM sleep.

We found there to be significant beta and theta activity in the ACC and the DLPFC (see Figure 3 and Figure 4 in the manuscript). The presence of rhythmic activity in the DLPFC is surprising since imaging studies suggest that the DLPFC is relatively inactive during REM sleep; the relative inactivity of the DLPFC, a structure thought to be important for executive functions, has been argued to be the cause of the illogical nature of our dreams. The high-frequency activity generated in the DLPFC presumably requires a fair amount of metabolic energy. Furthermore, the ACC and the DLPFC, two relatively distant structures, were coherent in the theta and beta bands, suggesting that the DLPFC may in fact be interacting with other structures during REM sleep. There was also significant theta and beta activity in the inferior frontal gyrus (see Figure 4 supplement in manuscript). There were not enough bipolar derivations to determine the nature of the rhythmic activity present in other frontal structures; note that the placement of the electrodes was determined strictly by clinical criteria. The frontal structures that were examined exhibited more high frequency activity (i.e., beta) than did temporal areas (see Figure 2 and Figure 5 in the manuscript). The difference in power between the various frontal structures cannot be determined using our data since power depends on the geometric arrangement of the neuronal process relative to a given electrode as well as the proximity to the electrode.

For subsequent analyses we used only electrodes that showed significant power in the band relevant to the analysis. However, this resulted in the elimination of very few electrodes. In the DLPFC no electrodes were eliminated from the theta band analyses (0/15) and 2 electrodes were eliminated from the beta band analyses (2/15). In the ACC no electrodes were eliminated from the beta band analyses (0/22) and 3 electrodes were eliminated from the theta band analyses (3/22).

We have now included the number of electrodes used for each of the relevant analyses in the manuscript.

To further address the reviewer's concerns, the following two paragraphs have been added to the section of the Methods entitled Identification of Contacts with Significant Theta and Beta Power. The relevant text is pasted below.

“We used this method rather than a comparison with awake activity since during awake conditions the calculated power spectrum would depend on the cognitive state and the task being performed. See Figure 3—figure supplement 2 for power spectrum comparisons between NREM sleep and REM sleep, which demonstrate that the spectral content is very different between the two states and show the clear distinction between spindling activity during NREM sleep and beta activity during REM sleep.”

“The authors limited their analysis to a potentially quite restricted subsample of the data. First, data from frontal electrodes were reduced to electrodes in the ACC and DLPFC, because beta and theta activity was most prominent in these regions. From these electrodes, only those were used for coherence analysis that showed a significant power increase in the frequency bands of interest. From these electrode pairs, only those that showed significant coherence were used for the subsequent power cross-covariance analysis, which was moreover only calculated for time windows with power in the top 25% within each REM episode. All of this data-driven sub-selection potentially bias the results towards positive findings. As an example, using only electrode pairs with high coherence for power cross-covariance can be problematic since these two measures are not independent (Lachaux, Rodriguez et al. 1999, Srinath and Ray 2014). Furthermore, it is not clear how much data (sleep epochs, electrodes, time windows, electrode pairs) were discarded in absolute numbers in each step (only the ratios of electrodes used based on power and coherence are given in Figure 3–Figure 6). Please report the number of recorded electrodes per subject including their location. Please give an idea on how many data were finally used at each analysis step.

We have now added details about the selection of data to be included in the analyses. The selection of the particular frontal areas used in the analyses was based on anatomical criteria and the requirement that each area has a sufficient number of electrodes across all subjects. As stated, since we used bipolar derivations using neighboring electrodes, we required that both electrodes be from the same anatomically defined region and that at least one of the electrode contacts be in the grey matter. These selection criteria resulted in 15 bipolar derivations in the DLPFC, 22 in the ACC, and 12 in the inferior frontal gyrus. These selection criteria allowed us to have confidence that the activity of a given derivation was local in nature and that it originated from the region of interest; if two contacts had been from two different regions it would not have been possible to determine the regional origin of the activity.

The reviewer correctly points out that we eliminated electrodes that did not show significant power in the theta or beta bands for subsequent analyses. However, this resulted in the elimination of relatively few electrodes. In the DLPFC no electrodes were eliminated from the theta band analyses (0/15) and 2 electrodes were eliminated from the beta band analyses (2/15). In the ACC no electrodes were eliminated from the beta band analyses (0/22) and 3 electrodes were eliminated from the theta band analyses (3/22). Therefore, for the coherence analysis we examined 34 ACC-DLPFC electrode pairs in the theta band (3 ACC electrodes eliminated) and 31 ACC-DLPFC electrode pairs in the beta band (2 DLPFC electrodes eliminated).

Cross-Covariance Calculations

Since the coherence measure indicated that there was a relationship between the time series of the electrode pairs, we were interested in determining whether on average a rhythm originated in one region or the other when there was ongoing rhythmic activity in both bipolar derivations. In other words, the primary motivation was to determine the time lag, not the amplitude, of the relationship. Since we were interested in electrode pairs whose time series had a temporal relationship, we restricted our analysis to pairs of electrodes whose coherence measure was significant. Also, to get a better estimate of the lag we restricted the analysis to periods when the power was high and employed the cross-covariance method as described in (Adhikari et al., 2010). We used 23 ACC-DLPFC bipolar electrode pair derivations for the beta band cross-covariance analysis and 26 ACC-DLPFC bipolar electrode pair derivations for the theta band cross-covariance analysis (we omitted 8 electrode pairs from the theta and beta analyses).

We calculated cross-covariance by first filtering the raw signal in the band of interest during the REM episodes and five minutes before and after the REM episodes to avoid edge effects. The instantaneous amplitude was then calculated using the Hilbert transform of the filtered signal; following this step we used the data from only the REM period. Using the calculated instantaneous amplitude the cross-covariance of the power (amplitude) of an ACC-DLPFC electrode pair was calculated for each two- second REM sleep window. For a given electrode the cross-covariance functions for all two-second windows were averaged to obtain an average cross-covariance function (Adhikari et al., 2010).

To address the reviewer’s concerns we have added details about the number of electrodes used in each step. We have also added the following sentence to the section entitled Power Cross-Covariance Estimates (in the Materials and,methods section).

*“*We used 23 ACC-DLPFC bipolar electrode pair derivations for the beta band cross- covariance analysis and 26 ACC-DLPFC bipolar electrode pair derivations for the theta band cross-covariance analysis (we omitted 8 electrode pairs from the theta and beta analyses).*”*

Activity Across Subjects

The examples of spectral data that we included in the manuscript are representative. For each example spectrogram from a single subject, we also included the average power spectrum (averaged across all subjects). Note that these average power spectra show significant power in the same bands in which the example spectrogram shows power. For example, the example spectrogram in Figure 3 in the manuscript shows activity in the theta and beta bands, and the average power spectrum in Figure 3 in the manuscript also shows power in the theta and beta bands. To give a better sense of the distribution of the power across the subjects and to address the reviewer’s concerns we have now added lines showing +/- the standard error for the power spectrum across the subjects. In Figure 3 we have added traces showing +/- the standard error for the power spectrum across the subjects. The included standard error traces for the ACC, DLPFC, and Middle Temporal Gyrus indicate that the observed beta and theta activity is consistent across the subjects. Note that in the instance of the inferior frontal gyrus, for which we provide data in the figure supplements, the standard error traces indicate that theta and beta activities are variable across the subjects.

*Coherence was tested for statistically significant differences using Monte Carlo simulations as well as the "confidence level provided by Chronux" (subsection “Coherence Estimates”). What is this confidence level based on? Apparently, both methods yielded "the same results" (Paragraph six Results section). Please elaborate, are the results indeed fully identical?*

Coherence Calculations II

We determined significance using two different methods. One method was to use the confidence level provided by Chronux. The confidence level is based on the distribution of the statistic (i.e., the coherence measure) given the null hypothesis of zero coherence (Bokil et al., 2010; Brillinger, 2001; Jarvis and Mitra, 2001). The distribution of the estimator is given by the following formula.

In the formula m is the degrees of freedom, which is equal to the number of 2-second intervals multiplied by the number of data tapers used in the calculation. See Brillinger (2001) for details regarding the asymptotic convergence of the coherence estimator under the hypothesis of zero coherence. The other method was via Monte Carlo simulations (2,000 shuffles). A shuffle was done by permuting the 2-second windows of one of the electrode pairs and then calculating the coherence. Using the distribution of coherences from the shuffles we determined the probability of getting the coherence value we got using our data. Both of these methods gave identical results in the sense that they identified the same set of electrodes as having significant beta activity and the same set of electrodes as having significant theta activity.

To address the reviewer's concerns and make these details clearer, we have amended the text in the Materials and methods section entitled Coherence Estimates. The relevant text is pasted below.

*“*We assessed significance using two methods: by using the confidence level provided by Chronux and by doing Monte Carlo simulations (2,000 shuffles). The confidence level provided by Chronux is based on distribution of the statistic (i.e., the coherence measure) given the null hypothesis of zero coherence (Bokil et al., 2010; Brillinger, 2001; Jarvis and Mitra, 2001). The distribution of the estimator is given by the following formula.

In the formula m is the degrees of freedom, which is equal to the number of 2-second intervals multiplied by the number of data tapers used in the calculation. See Brillinger (2001) for details regarding the asymptotic convergence of the coherence estimator under the hypothesis of zero coherence. For the Monte Carlo simulations, a shuffle was done by permuting the 2-second windows of one of the electrode pairs and then calculating the coherence. Using the distribution of coherences from the shuffles we determined the probability of getting the coherence value we got using our data. Both methods identified the same set of DLPFC-ACC electrode derivation pairs as having significant (p <.05, Bonferroni corrected for number of electrode pairs) beta or theta coherence.”

How did the authors deal with the influence of the number of trials for the estimation of coherence (e.g. Maris, Schoffelen, Fries 2007). Was the number of observations equalized across REM sleep periods?

Note that we did not calculate any coherence differences as in (Maris et al., 2007). For a given electrode pair in a given subject we divided all the REM episodes into 2-second bins and then made a single coherence calculation. Because we were interested not in differences but rather in the significance relative to a null hypothesis we did not need to adjust for the number of observations (the number of observations is taken into account by the confidence level). We have clarified this point by amendments to the text, which are detailed in the responses to 1c-i and 1c-ii.

*In the subsection “Coherence Estimates”, the calculation of coherence reads as if the coherence between two electrodes was calculated for each window and then was averaged. However, since calculating coherence for a single trial (here: 2-s time window) will always result in a coherence of 1 (as well as their average), I assume that the cross-spectra were calculated for each time window and then used to calculate coherence (not "average coherence") across time windows. Please clarify the procedure”.*

As the reviewer points out our description of the coherence calculation needs clarification.

Coherence Calculations I

Coherence calculations were made using the Chronux toolbox. Coherence was examined for 34 ACC-DLPFC bipolar electrode derivations in the theta band (3 ACC electrodes were eliminated because theta power was not significant) and 31 ACC- DLPFC electrode pairs in the beta band (2 DLPFC electrodes were eliminated because beta power was not significant). For each subject REM episodes were divided into 2- second windows. All of the 2-second windows were used to make a single coherence calculation for that subject using the coherence formula employed by Chronux (see equations 7.77 and 7.80 in Mitra and Bokil (2008)). For each frequency band, the time- bandwidth was set such that the coherence estimates for the center band (e.g., 6 Hz for theta) integrated the entire band of interest.

To address the reviewer's concerns, we have modified the Methods section entitled Coherence Estimates. The relevant text is pasted below.

“Coherence was examined for 34 ACC-DLPFC electrode pairs in the theta band (3 ACC electrodes were eliminated because theta power was not significant) and 31 ACC- DLPFC electrode pairs in the beta band (2 DLPFC electrodes were eliminated because beta power was not significant). We estimated the coherence during REM sleep by first dividing all REM sessions of a subject into 2-second windows. All 2-second intervals of data were included in a single coherence calculation for that subject using the coherence formula employed by Chronux (see equations 7.77 and 7.80 in Mitra and Bokil (2008)). The time-bandwidth was set such that the coherence estimate for the center band (e.g., 6 Hz for theta) integrated the entire band of interest.“

*Please state in significantly clearer way how you conducted the analyses, what was compared, or how the statistical significance was obtained. Bin length, definition of regions, data selection (if any), procedures of a coherence analysis, cross-covariance analysis, and data visualization (Figure 2–Figure 6 each show spectral data from only one bipolar derivation or electrode pair. How representative are these data? Is there a way to present data aggregated from more than one electrode / subject?), etc. are unclear. In addition, please clarify whether you compared REM sleep vs. NREM sleep or Wake stage.*

We have now included bin lengths and data selection criteria in each of the relevant sections. Details about the definition of regions can be found in the response titled Identification of Electrode Localization. Details about coherence analyses can be found in the responses sections titled Coherence Calculations I and Coherence Calculations II. Details about cross-covariance analyses can be found in the response section titled Cross-Covariance Calculations.

The data are representative across the subjects; we have modified some of the figures to give a better sense of the consistency across the subjects. Please see the section entitled Activity Across Subjects for more details. We have also provided details about the comparisons that were made and how significances were obtained; these details are in multiple sections, but most notably in the response section titled Determination of Significant Theta and Β Activity.

*2) All reviewers agreed that discussion about REM sleep and memory consolidation are too far-reaching based on the present data, particularly because it is not shown that the obtained effects are specific to REM sleep.*

*Please revise the relevant statements so that the manuscript would not go too far beyond the current data.*

We have substantially modified and reduced our discussion of REM sleep and memory. We have made it clear that our claims are hypotheses and we have added the necessary caveats. Since the functional role of REM sleep remains largely a mystery, we feel that some hypotheses about the putative functional role of the rhythmic activity we observed are warranted in the discussion to motivate future studies. Of course, we are open to further modifications if the changes we have made are unsatisfactory. We have rewritten this section and pasted it below in response to one of the reviewer comments.

*“The function of REM in procedural memory consolidation is questionable as the result is not found in many studies. Recent studies seem to suggest that REMs role in emotional memory consolidation is limited (e.g., see Payne et al., 2015) and perhaps reflects emotional processing without consolidation (see Baran et al., 2012). Keeping in mind that the present data contain no cognitive measures, it cannot be stated "These findings, in conjunction with previous work on REM sleep, suggest a system for the consolidation of memories during REM sleep." It seems that much of the discussion is focused on just this point – the REM role in consolidation – which I see no clear support for in the present data. Sure they support some cognitive role of REM but is it necessarily emotional or procedural memory consolidation? That link is absent”.*

To address the reviewer’s concerns we have substantially modified the Discussion section and any claims about the functional role of REM sleep in the manuscript. The new Discussion section is pasted below.

*“*What functional role sleep plays has yet to be resolved. There is a large body of evidence suggesting that NREM sleep may play an important role in memory consolidation (Rasch and Born, 2013), but whether or not REM sleep also plays a role in memory consolidation remains unclear (Dudai et al., 2015; Rasch and Born, 2013).[…] Furthermore, emotional information from the limbic system conveyed via the theta band may bias the processing and consolidation of procedural memories with high emotional content (Popa et al., 2010). Note that many different functions have been attributed to the ACC (Paus, 2001), so its activation during REM sleep does not necessarily mean that it serves the function of emotional or procedural memory consolidation during REM sleep.”

“The authors speculate that activation of the ACC during REM sleep relates to consolidation motor learning. However, there are many functions in which the ACC is suggested to play roles. Mere activation of the ACC during REM sleep may leave other possibilities than its involvement in consolidation of motor learning”.

As the reviewer points out activation of the ACC during REM sleep could serve some other role than the consolidation of motor learning.

To address the reviewer's concerns and make these details clearer, we have added the following to the text in the Discussion section entitled Functional Implications. The relevant text is pasted below.

“Note that many different functions have been attributed to the ACC (Paus, 2001), so its activation during REM sleep does not necessarily mean that it serves the function of emotional or procedural memory consolidation during REM sleep.”